# Yap suppresses T-cell function and infiltration in the tumor microenvironment

**Eleni Stampouloglou**[1], **Nan Cheng**[1], **Anthony Federico**[2,3], **Emily Slaby**[4], **Stefano Monti**[2,3], **Gregory L. Szeto**[4,5,6,7]*, **Xaralabos Varelas**[1]*

**1** Department of Biochemistry, Boston University School of Medicine, Boston, Massachusetts, United States of America, **2** Division of Computational Biology, Department of Medicine, Boston University School of Medicine, Boston, Massachusetts, United States of America, **3** Bioinformatics Program, Boston University, Boston, Massachusetts, United States of America, **4** Department of Chemical, Biochemical, and Environmental Engineering, University of Maryland Baltimore County, Baltimore, Maryland, United States of America, **5** Center for Biomedical Engineering and Technology, University of Maryland School of Medicine, Baltimore, Maryland, United States of America, **6** Marlene and Stewart Greenebaum Comprehensive Cancer Center, University of Maryland, Baltimore, Maryland, United States of America, **7** Translational Center for Age-Related Disease and Disparities, University of Maryland Baltimore County, Baltimore, Maryland, United States of America

* xvarelas@bu.edu (XV); gszeto@umbc.edu (GLS)

**Data Availability Statement:** All relevant data are noted within the paper and included as Supporting Information files. RNA-seq data have been deposited in NCBI's Gene Expression Omnibus and

## Abstract

A major challenge for cancer immunotherapy is sustaining T-cell activation and recruitment in immunosuppressive solid tumors. Here, we report that the levels of the Hippo pathway effector Yes-associated protein (Yap) are sharply induced upon the activation of cluster of differentiation 4 (CD4)-positive and cluster of differentiation 8 (CD8)-positive T cells and that Yap functions as an immunosuppressive factor and inhibitor of effector differentiation. Loss of Yap in T cells results in enhanced T-cell activation, differentiation, and function, which translates in vivo to an improved ability for T cells to infiltrate and repress tumors. Gene expression analyses of tumor-infiltrating T cells following Yap deletion implicates Yap as a mediator of global T-cell responses in the tumor microenvironment and as a negative regulator of T-cell tumor infiltration and patient survival in diverse human cancers. Collectively, our results indicate that Yap plays critical roles in T-cell biology and suggest that Yap inhibition improves T-cell responses in cancer.

## Introduction

Cluster of differentiation 8-positive (CD8$^+$) and cluster of differentiation 4-positive (CD4$^+$) T cells are central players in the adaptive immune system. T cells elicit targeted, antigen-specific responses for direct killing of an infected or transformed cell, shaping and regulating the immune response in host defense [1]. Most mature T cells circulate in a resting, naïve state, and upon cognate antigen recognition, T cells become activated, proliferate clonally, and differentiate into effector T cells. Naïve CD8$^+$ T cells differentiate into cytotoxic T cells, while CD4$^+$ T cells differentiate into an array of different types of helper (i.e., T helper cell type 1 [Th1], Th2, Th17) or regulatory T cells (Tregs) depending on microenvironmental cues [2].

are accessible through GEO Series accession number GSE139883. Raw data for Fig S1K have been deposited in FLOWRepository (https://flowrepository.org/) (Repository ID: FR-FCM-Z2D5).

**Funding:** XV was funded by a grant from the NIH National Heart Lung and Blood Institute (R01HL124392). GLS is funded in part by the UMGCC P30 grant under award number P30CA134274 from the National Cancer Institute, NIH. The funders had no role in study design, data collection and analysis, decision to publish, or preparation of the manuscript.

**Competing interests:** The authors have declared that no competing interests exist.

**Abbreviations:** ACC, adrenocortical carcinoma; BLCA, bladder urothelial carcinoma; BRCA, breast invasive carcinoma; CAR-T, chimeric antigen receptor T cell; CD, cluster of differentiation; CESC, cervical squamous cell carcinoma and endocervical adenocarcinoma; CHOL, cholangiocarcinoma; cKO, conditional knockout; COAD, colon adenocarcinoma; DEG, differentially expressed gene; DLBC, lymphoid neoplasm diffuse large B-cell lymphoma; DLN, draining lymph node; DN, double negative; DP, double positive; ESCA, esophageal carcinoma; EYFP, enhanced yellow fluorescent protein; Foxp3, forkhead box protein 3; GAPDH, glyceraldehyde 3-phosphate dehydrogenase; GATA3, GATA binding protein 3; GBM, glioblastoma multiforme; GDC, Genomic Data Commons; GEO, Gene Expression Omnibus; GSVA, Gene Set Variation Analysis; HNSC, head and neck squamous cell carcinoma; IFNγY, interferon gamma; IL-4, interleukin 4; JAK-STAT, Janus kinase-signal transducers and activators of transcription; KICH, kidney chromophobe; KIRC, kidney renal clear cell carcinoma; KIRP, kidney renal papillary cell carcinoma; LATS1, Large tumor suppressor kinase 1; LGG, brain lower grade glioma; LIHC, liver hepatocellular carcinoma; LLC, Lewis lung carcinoma; loxP, locus of X-over P1; LSL, lox-stop-lox; LUAD, lung adenocarcinoma; LUSC, lung squamous cell carcinoma; MAPK, mitogen-activated protein kinase; MESO, mesothelioma; MFI, median fluorescence intensity; MHC, major histocompatibility complex; MHCI, major histocompatibility complex class I; MST1, mammalian sterile 20 like 1; mTOR, mammalian target of rapamycin; NCBI, National Center for Biotechnology Information; NFAT, nuclear factor of activated T cells; NFκB, nuclear factor kappa-light-chain-enhancer of activated B cells; Nur77, nuclear receptor 77; OV, ovarian serous cystadenocarcinoma; PAAD, pancreatic adenocarcinoma; PCPG, pheochromocytoma and

Each phenotype is defined by expression of signature transcription factors and effector cytokines leading to distinct functions [3].

T-cell activation also up-regulates negative feedback mechanisms, such as inhibitory receptors, which minimize pathogenic inflammation and autoimmunity [1,4]. This network of immunosuppressive factors is frequently co-opted in chronic infections and cancer, leading to terminally differentiated and exhausted T cells that lose effector function and ability to infiltrate disease sites [5]. The finding that revitalization of exhausted, dysfunctional T cells can restore the immune response has revolutionized cancer therapy with the use of checkpoint inhibition [6]. Chimeric antigen receptor T cells (CAR-Ts), engineered for enhanced antigen recognition and costimulation, also demonstrate promising clinical efficacy [7–9]. However, both immunotherapies are effective for only a fraction of patients [10–14]. Major challenges to extending the efficacy of immunotherapy to more cancer patients include sustaining T-cell activation and achieving T-cell infiltration in the immunosuppressive microenvironment of solid tumors [15–24]. Improved understanding of mechanisms controlling T-cell differentiation and function is critical to overcoming these barriers.

Yes-associated protein (Yap) is a key effector of the Hippo signaling pathway, directing transcriptional programs that control stem cell biology by integrating microenvironmental and cell-intrinsic cues [25]. Yap controls signals that mediate cell death and survival, proliferation, and cell fate determination, while dysregulated Yap activity contributes to disease, most notably cancer [26]. While the dynamics of Yap regulation coupled with differentiation are well characterized in stem cells and tissue-specific progenitor cells, less is known about the role of Yap in T cells. The Hippo pathway has been implicated in coupling CD8$^+$ T-cell clonal expansion to terminal differentiation through up-regulation of the Large tumor suppressor kinase 1 (LATS1) kinase and Yap degradation [27]. The Hippo pathway kinases mammalian sterile 20-like 1/2 (MST1/2) have also been implicated in thymocyte egress and antigen recognition, lymphocyte polarization, adhesion and trafficking, survival, differentiation and proliferation [28–36]. Furthermore, an immune-cell–intrinsic role for Yap in CD4$^+$ T cells has also been described, with Yap potentiating transforming growth factor beta (TGFβ) signaling responses that direct Treg function [37].

In this study, we investigated the role of Yap in primary CD4$^+$ and CD8$^+$ T-cell responses—namely, activation, proliferation, and differentiation into effector T-cell subsets—and how Yap might affect T-cell responses in cancer. We observed that Yap levels are elevated upon T-cell activation and that conditional deletion of the *Yap* gene in CD4$^+$ and CD8$^+$ T cells enhanced their activation and differentiation potential. These phenotypes translated in vivo to reduced growth of B16F10 melanoma and Lewis lung carcinoma (LLC) lung cancer tumors in mice with Yap-deleted T cells, with these mice showing notably increased T-cell infiltration into tumors. Using adoptive T-cell transfer experiments, we observed that Yap-deleted polyclonal CD8$^+$ T cells have an intrinsic ability to infiltrate tumors with higher efficiency. RNA sequencing (RNA-seq) analyses of Yap-deleted tumor-infiltrating lymphocytes (TILs) revealed up-regulation of key signals important for CD4$^+$ and CD8$^+$ T-cell activation, differentiation, and function. Notably, we found that Yap-regulated gene expression changes were tumor specific, as we observed minimal gene expression changes in lymphocytes isolated from tumor-draining lymph nodes (TDLNs). Yap-regulated gene expression signatures from TILs correlated with T-cell infiltration and patient survival across multiple human cancers in The Cancer Genome Atlas (TCGA), including melanoma and lung cancer, consistent with our mouse studies. Our study highlights Yap as a broad suppressor of CD4$^+$ and CD8$^+$ T-cell activation and function and a key regulator of T-cell tumor infiltration and survival in cancer immunotherapy patients.

paraganglioma; PMA, phorbol 12-myristate 13-acetate; PRAD, prostate adenocarcinoma; READ, rectum adenocarcinoma; RLE, relative log expression; RNA-seq, RNA sequencing; RORγt, RAR-related orphan receptor γt; SARC, sarcoma; SKCM, skin cutaneous melanoma; SM, semimature; SP, single positive; STAD, stomach adenocarcinoma; T-BET, T-box protein expressed in T cells; TCGA, The Cancer Genome Atlas; TCR, T cell receptor; TDLN, tumor-draining lymph node; TEAD, TEA domain family member; TGCT, testicular germ cell tumors; TGFβ, transforming growth factor beta; Th1, T helper cell type 1; THCA, thyroid carcinoma; THYM, thymoma; TIL, tumor-infiltrating lymphocyte; TIMER, Tumor Immune Estimation Resource; Treg, regulatory T cell; UCEC, uterine corpus endometrial carcinoma; UCS, uterine carcinosarcoma; UVM, uveal melanoma; WT, wild-type; Yap, Yes-associated protein.

## Results

### Yap inhibits CD4$^+$ and CD8$^+$ T-cell activation

To study the role of Yap in T cells, we started by analyzing Yap levels in isolated mouse primary CD4$^+$ and CD8$^+$ T cells that were either unstimulated or activated with anti-CD3/CD28 coated beads across a range of time points up to 24 hours. We observed rapid induction of Yap protein in both CD4$^+$ and CD8$^+$ T cells upon in vitro activation (Fig 1A and 1B). This observation contrasted with prior reports that Yap is exclusively expressed in Tregs or CD8$^+$ T cells cultured under specific conditions [27,37], which prompted us to systematically explore roles for Yap in CD4$^+$ and CD8$^+$ T-cell activation and function. To gain insight into the roles of Yap in T cells, we generated a mouse model in which Yap deletion and enhanced yellow fluorescent protein (EYFP) expression are induced under the control of the CD4 promoter (Yap-locus of X-over P1 [loxP]/loxP; lox-stop-lox [LSL]-EYFP; CD4-Cre, herein referred to as Yap-conditional knockout [cKO]). Yap expression was efficiently reduced in both CD4$^+$ and CD8$^+$ T cells in Yap-cKO mice, which was expected given the activity of this Cre model at the CD4$^+$CD8$^+$ double-positive (DP) stage of T-cell development (S1A and S1B Fig), and cells were also efficiently marked by EYFP expression (S1C and S1D Fig). No systemic defects or gross phenotypes were observed in Yap-cKO mice housed in a barrier facility.

We next measured the surface levels of activation markers CD44, CD69, and CD25 in Yap-cKO T cells [38–41]. CD4$^+$ and CD8$^+$ T cells were isolated from wild-type (WT) and Yap-cKO mouse spleens, activated by increasing concentrations of plate-bound anti-CD3, and levels of activation markers were measured 72 hours later. We found that Yap-cKO CD4$^+$ and CD8$^+$ T cells were more sensitive to anti-CD3 stimulation compared to WT cells, showing increased expression of CD44 (Fig 1C and 1D), CD25 (Fig 1E and 1F), and CD69 (S1E and S1F Fig). CD44 and CD25 in CD4$^+$ and CD8$^+$ T cells exhibited significantly higher expression under all stimulation conditions tested, with CD69 being significantly up-regulated only after stimulation with intermediate anti-CD3 concentrations (0.25 and 0.5 μg/mL for CD8$^+$ T cells, and 0.125 μg/mL for CD4$^+$ T cells). CD4$^+$ and CD8$^+$ T-cell proliferation was also measured, but no significant differences were observed between WT and Yap-cKO cells 72 hours post stimulation (S1G and S1H Fig). These data indicate that Yap plays an inhibitory role in T-cell activation and that loss of Yap enhances the sensitivity of CD4$^+$ and CD8$^+$ T cells to T cell receptor (TCR) signaling.

Yap functions as a transcriptional regulator, with the best characterized roles being the regulation of the TEA domain family member (TEAD) transcriptional factors. We observed a substantial increase in the expression of TEAD1 and TEAD3 in CD4$^+$ T cells (Fig 1G) and TEAD1 in CD8$^+$ T cells upon activation (Fig 1H). These increases in TEAD expression prompted us to test the effects of the small-molecule drug verteporfin (also known as Visudyne)—which has been reported to inhibit Yap-mediated activation of TEADs [42]—on T-cell activation and proliferation. Verteporfin treatment of CD4$^+$ and CD8$^+$ T cells isolated from WT mice increased levels of early activation markers CD71 (Fig 1I and 1J) and CD69 (S1I and S1J Fig) in a concentration-dependent manner. Similar to our observations with Yap-cKO cells, verteporfin treatment did not significantly affect proliferation of T cells isolated from WT mice 3 days post stimulation (S1K Fig). These data suggest that Yap-regulated transcription plays a suppressive function following T-cell activation.

### Yap inhibits CD4$^+$ T-cell differentiation

Upon encountering their cognate antigen and receiving appropriate costimulation, naïve CD4$^+$ T cells become activated and can differentiate into several functionally diverse subsets,

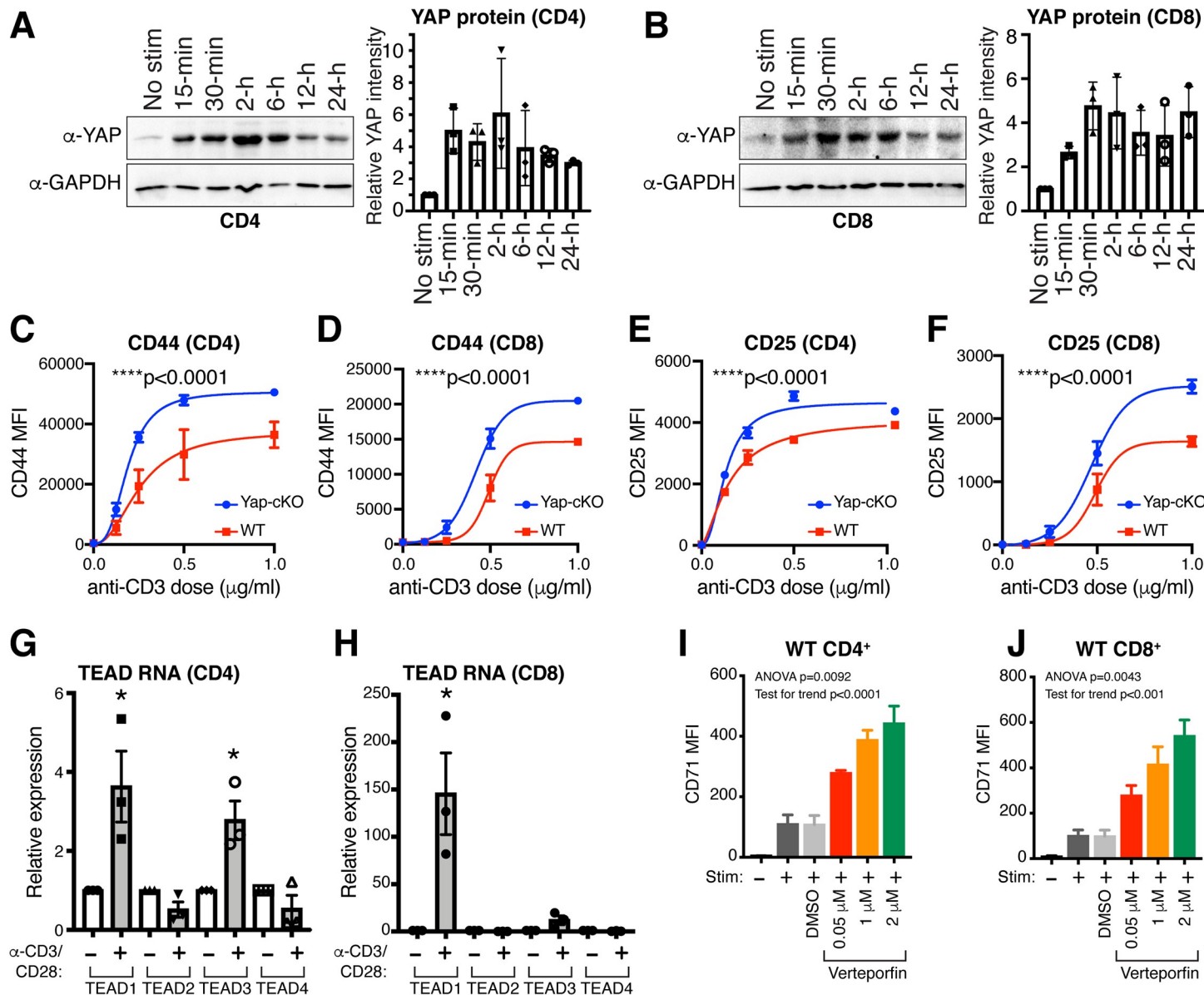

**Fig 1. Yap expression is induced upon T-cell activation resulting in suppression of T-cell activation.** WT and Yap-cKO CD4+ and CD8+ T cells were isolated from mouse spleens and stimulated with anti-CD3 and anti-CD28 coated magnetic beads to test expression of Yap protein and TEAD1–4 mRNA. T-cell activation was tested using increasing concentrations of plate-bound anti-CD3 with soluble anti-CD28 in Yap-cKO and WT T cells. Activation marker expression was also tested in WT CD4+ and CD8+ T cells treated with increasing concentrations of verteporfin under IL-2, and CD3/CD28 stimulation. Statistical differences were determined by using an F test to identify differences between nonlinear curve fits (C–F), unpaired two-sample $t$ test (G–H), or one-way repeated-measures ANOVA with post hoc test for linear trend with increasing verteporfin (I–J). The underlying data for the graphs in this figure can be found in S1 Data and for the immunoblots in S1 Raw Images. (A) Yap protein levels in CD4+ T cells isolated from WT mouse spleens at various time points following CD3/CD28 stimulation. (B) Yap protein levels in CD8+ T cells isolated from WT mouse spleens at various time points following CD3/CD28 stimulation. (C) CD44 expression by flow cytometry on WT and Yap-cKO CD4+ T cells 72 hours post CD3/CD28 stimulation ($n = 2$–3 per dose/group). (D) CD44 expression by flow cytometry on WT and Yap-cKO CD8+ T cells 72 hours post CD3/CD28 stimulation ($n = 2$–3 per dose/group). (E) CD25 expression by flow cytometry on WT and Yap-cKO CD4+ T cells 72 hours post CD3/CD28 stimulation ($n = 2$–3 per dose/group). (F) CD25 expression by flow cytometry on WT and Yap-cKO CD8+ T cells 72 hours post CD3/CD28 stimulation ($n = 2$–3 per dose/group). (G) TEAD1–4 mRNA expression in CD4+ T cells 24 hours post CD3/CD28 stimulation ($n = 3$/group). (H) TEAD1–4 mRNA expression in CD8+ T cells 24 hours post CD3/CD28 stimulation ($n = 3$/group). (I) CD71 expression on WT CD4+ T cells 72 hours post IL-2 and CD3/CD28 stimulation and increasing concentration of verteporfin ($n = 4$/group). (J) CD71 expression on WT CD8+ T cells 72 hours post IL-2 and CD3/CD28 stimulation and increasing concentration of verteporfin ($n = 4$/group). CD, cluster of differentiation; cKO, conditional knockout; GAPDH, glyceraldehyde 3-phosphate dehydrogenase; IL-2, interleukin 2; MFI, median fluorescence intensity; No Stim, no stimulation; Stim, stimulation with IL-2 and anti-CD3/CD28; TEAD, TEA domain family member; WT, wild type; Yap, Yes-associated protein.

including Th1, Th2, Th17, and Tregs. These subsets contribute to protective immunity or immunopathology depending on microenvironmental signals [3]. Signature transcription factors and effector cytokines define each subset: Th1 is defined by expression of T-box protein expressed in T cells (T-BET) and interferon gamma (IFNγ), Th2 by GATA binding protein 3 (GATA3) and interleukin 4 (IL-4), Th17 by RAR-related orphan receptor γt (RORγt) and IL-17, and Treg by forkhead box protein 3 (Foxp3). Given the wealth of evidence for Yap playing key roles in stem cell regulation, we hypothesized that deletion of Yap alters the differentiation potential of CD4$^+$ T cells. To test this hypothesis, we isolated naïve CD4$^+$ T cells from WT and Yap-cKO mice and cultured them under Th1-, Th17-, Th2-, and Treg-polarizing conditions [43,44]. We found that Yap-deleted CD4$^+$ T cells showed significantly enhanced Th1 (Fig 2A), Th17 (Fig 2B), Th2 (Fig 2C), and Treg (Fig 2D) differentiation compared to WT cells, demonstrated by increased intracellular IFNγ, IL-17, GATA3, and Foxp3 expression, respectively. We observed higher Foxp3 induction at lower concentrations of TGFβ in Yap-cKO cells compared to WT controls, indicating that Yap-deleted cells were more responsive to Treg differentiation conditions. Collectively, these observations are distinct from prior findings that suggested that Yap functions only in Tregs [37] and implicate Yap as an inhibitor of CD4$^+$ T-cell activation and differentiation into Th1, Th17, Th2, and Treg fates.

## Yap deletion does not alter T-cell development or output in the thymus

Next, we compared thymic populations from WT and Yap-cKO mice to determine whether alterations in T-cell development may play a role in the observed functional changes. During maturation in the thymus, precursor CD4$^+$CD8$^+$ DP thymocytes go through positive selection to identify thymocytes expressing TCRs that can functionally bind peptide–major histocompatibility complex (MHC) complexes in the cortex [45]. Positively selected thymocytes become single positive (SP) and migrate from the cortex to the medulla, where self-reactive T cells die during negative selection, while surviving T cells exit into systemic circulation. Negative selection eliminates T cells with high affinity for self-peptide/self-MHC to reduce the potential for autoimmunity. Total cell counts in WT and Yap-cKO mice were determined across the continuum of thymocyte maturation, including double-negative (DN) (CD4$^-$CD8$^-$), DP (CD4$^+$CD8$^+$), and SP (CD4$^+$ or CD8$^+$) T cells (Fig 3A). No significant changes were observed in total cell count at any stage, suggesting that output from positive and negative selection processes were not broadly changed in Yap-cKO mice compared to WT.

Next, we analyzed the developmental progression of thymocytes through positive selection in greater detail using TCRβ and CD69 as markers. Yap-cKO mice compared to WT mice had no significant changes in the percentage of preselection (TCRβ$^-$CD69$^-$), post-TCR engagement (TCRβ$^-$CD69$^+$), or mature (TCRβ$^+$CD69$^-$) thymocytes (Fig 3B). There were significantly more Yap-cKO thymocytes in the post–positive-selection stage (TCRβ$^+$CD69$^+$) compared to WT mice (approximately 1% increase in total cells). However, thymocyte frequency in the next maturation stage (TCRβ$^+$CD69$^-$) was similar between Yap-cKO and WT mice, indicating no significant change in output of mature TCRβ$^+$CD69$^-$ cells. Overall, these observations suggest comparable selection output and TCRβ selection between Yap-cKO and WT mice.

Positively selected SP thymocytes (TCRβ$^+$CCR7$^+$) can be further functionally defined using markers CD69 and MHC class I (MHCI; H-2Kb) [46]. We observed no significant differences between Yap-cKO and WT mouse thymocyte frequency in CD69$^+$MHCI$^-$ semimature (SM) (Fig 3C), CD69$^+$MHCI$^+$ mature 1 (M1) (Fig 3D), and CD69$^-$MHCI$^+$ mature 2 (M2) (Fig 3E) stages. CD69 and nuclear receptor 77 (Nur77) are both markers of recent TCR engagement and rapidly degrade after signal removal [47,48]. However, CD69 also controls thymocyte

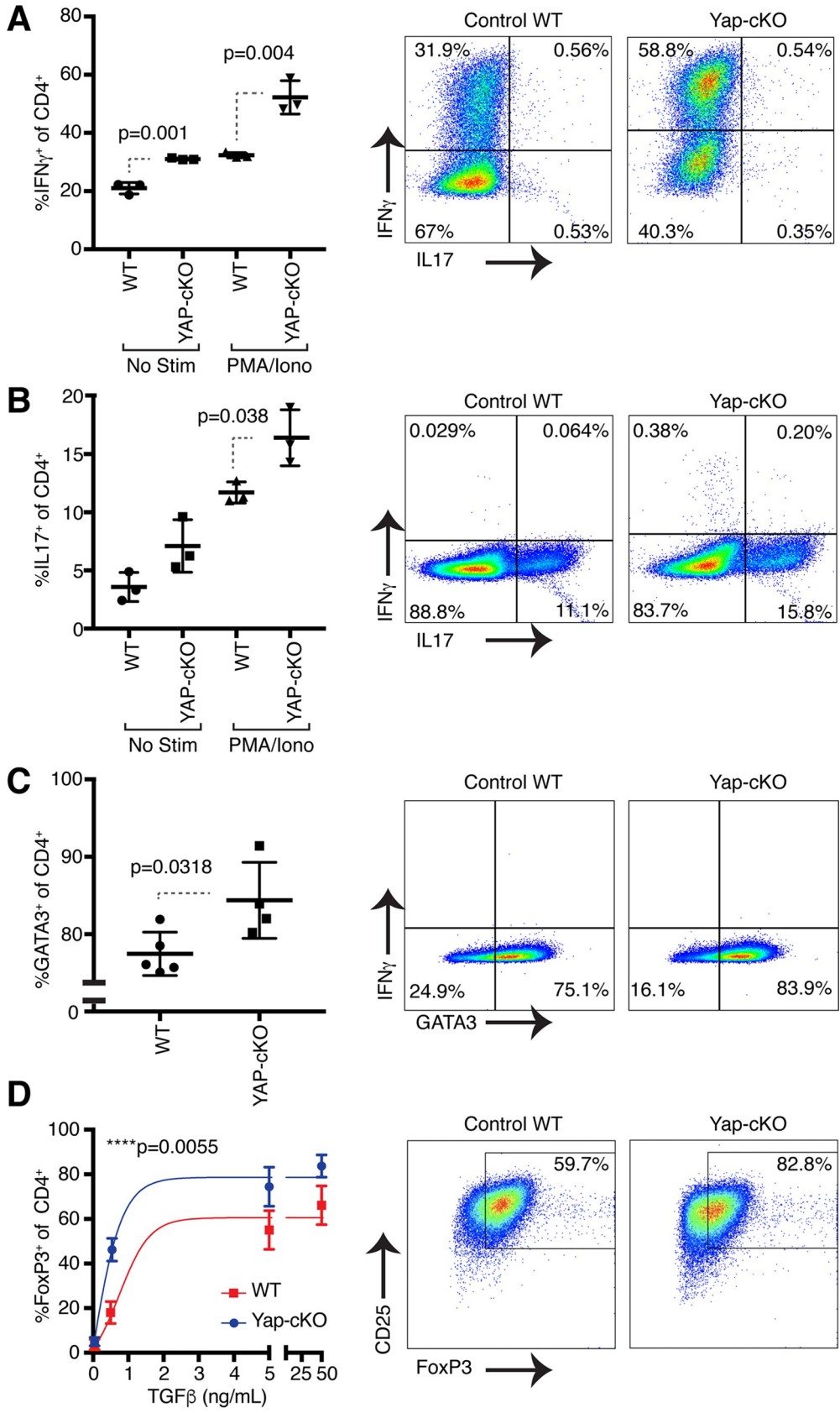

**Fig 2. Deletion of Yap in CD4$^+$ T cells results in increased IFNγ, IL-17, GATA3, and Foxp3 expression under Th1-, Th17-, Th2-, and Treg-polarizing conditions, respectively.** Naïve CD4$^+$ T cells from WT and Yap-cKO mice were isolated using magnetic beads. WT and Yap-cKO T cells were cultured under Th1-, Th17-, Th2-, or Treg-polarizing conditions for 5 days in the presence of CD3 and CD28 antibodies. On day 5, IFNγ, IL-17, GATA3, and Foxp3 expression were measured using flow cytometry. Statistical differences were determined by using a Student $t$ test (A–C) or an F test to identify differences between nonlinear curve fits (D). The underlying data for the graphs in this figure can be found in S2 Data. (A) IFNγ and IL-17 expression in WT and Yap-cKO CD4$^+$ T cells under Th1-polarizing conditions ($n$ = 3/group). (B) IFNγ and IL-17 expression in WT and Yap-cKO CD4$^+$ T cells under Th17-polarizing conditions ($n$ = 3/group). (C) IFNγ and GATA3 expression in WT and Yap-cKO CD4$^+$ T cells under Th2-polarizing conditions ($n$ = 3/group). (D) CD25 and Foxp3 expression in WT and Yap-cKO CD4$^+$ T cells under Treg-polarizing conditions ($n$ = 3–6 per dose/group). CD, cluster of differentiation; cKO, conditional knockout; Foxp3, forkhead box protein 3; GATA3, GATA binding protein 3; IFNγ, interferon gamma; Iono, ionomycin; IL-17, interleukin 17; No Stim, no stimulation; PMA, phorbol 12-myristate 13-acetate; TGFβ, transforming growth factor beta; Th, T helper cell type; Treg, regulatory T cell; WT, wild type; Yap, Yes-associated protein.

emigration and exit from the medulla, while Nur77 is more specifically involved in negative selection [49–51]. In this role, Nur77 controls T-cell survival during negative selection to elim- inate any T cells with high affinity binding to self-peptide–MHC and can induce apoptosis in immature thymocytes [52,53]. We therefore compared Nur77 levels in Yap-cKO with WT in DP and SP thymocytes to assess negative selection. Our analysis showed no increase in Nur77 expression in Yap-cKO mice compared to WT (Fig 3F), suggesting that Yap-cKO thymocytes do not receive prolonged TCR signaling relative to WT cells. Altogether, our data from thymic populations suggest that Yap deletion does not significantly change the number or percentage of cells during thymocyte development or thymic output.

## Deletion of Yap in T cells promotes T-cell infiltration into solid tumors and blocks tumor growth

Having observed that Yap inhibits T-cell activation and differentiation in vitro, we decided to test the effect of Yap deletion in T cells following an immune challenge in vivo. For this, we decided to test antitumor T-cell responses using the B16F10 tumor model, which was chosen because of the associated poorly immunogenic phenotype and the highly immunosuppressive microenvironment that leads to low T-cell infiltration [54–57]. Yap deletion in T cells resulted in superior antitumor immunity, as evidenced by the significant delay in tumor growth in Yap-cKO compared to WT mice (Fig 4A and 4B), consistent with prior observations [37]. We observed similar reduced growth of subcutaneous LLC tumors in Yap-cKO mice (Fig 4C and 4D), suggesting a general role for Yap in antitumor T-cell responses.

Given the strong correlation between CD8$^+$ T-cell tumor infiltration and patient survival, as well as patient responses to immunotherapy [15–19,58–62], we investigated the extent of T- cell infiltration in tumors that developed in Yap-cKO versus WT mice. Immunofluorescence microscopy analysis revealed that tumors that developed in Yap-cKO mice were significantly more infiltrated with CD8$^+$ T cells at both the tumor center and tumor edge compared to WT mice (Fig 4E). Flow cytometry analysis revealed more Yap-cKO CD4$^+$ and CD8$^+$ T cells infil- trating tumors compared to WT counterparts (Fig 4F and 4H).

To directly address whether Yap-deleted T cells have an increased ability to infiltrate tumors, we isolated polyclonal CD8$^+$ T cells from Yap-cKO and WT mice and directly com- pared tumor-infiltrating capacity in adoptive T-cell transfer experiments in WT mice inocu- lated with B16F10 tumor cells (illustrated in Fig 4I). We isolated tdTomato$^+$ CD8$^+$ T cells from WT mice and EYFP$^+$ CD8$^+$ T cells from Yap-cKO mice and mixed them at a 1 to 1 ratio. Cell mixtures were intravenously injected into WT mice the same day as subcutaneous injection of B16F10 cells. Absolute numbers of tdTomato$^+$ (WT) and EYFP$^+$ (Yap-cKO) CD8$^+$ T cells

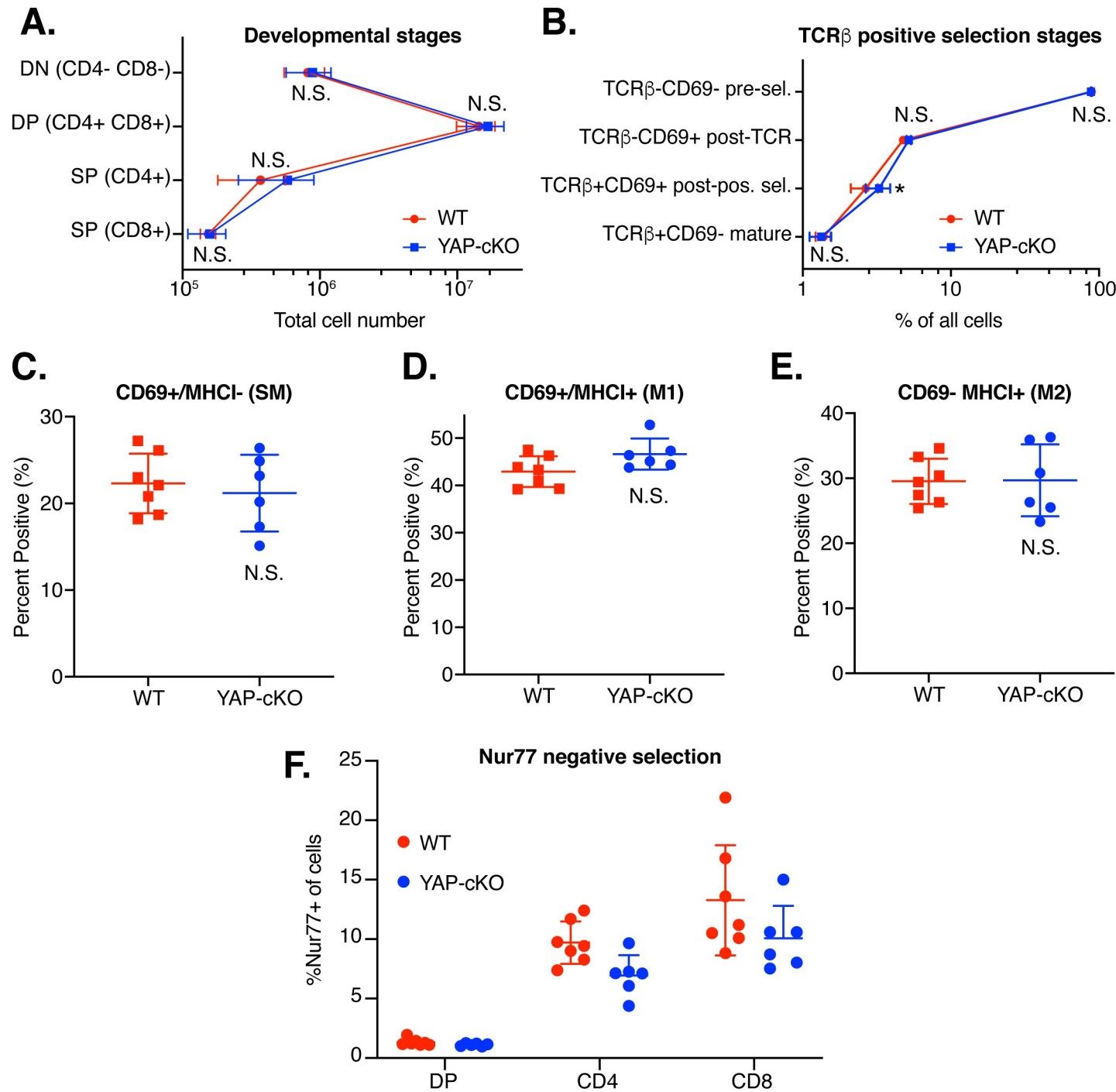

**Fig 3. Thymocyte development is similar between WT and Yap-cKO T cells.** (A) Total thymocytes were isolated from thymuses of WT and Yap-cKO mice. Cells were stained and analyzed by flow cytometry for coreceptor maturation (A), positive selection (B), medullary maturation (C–E), and negative selection (F). Statistical differences were determined using two-way ANOVA followed by Sidak's pairwise multiple comparisons tests. The underlying data for the graphs in this figure can be found in S3 Data. (A) Absolute numbers of thymocytes in different maturation stages: DN, DP, and SP for CD4 and CD8 coreceptor expression ($n$ = 6–7 mice per group). (B) Frequency of thymocytes progressing through positive selection determined by TCRβ and CD69 expression ($n$ = 6–7 mice per group). (C) Frequency of TCRβ⁺CCR7⁺ SP thymocytes in SM stage (CD69⁺MHCI⁻) ($n$ = 6–7 mice per group). (D) Frequency of TCRβ⁺CCR7⁺ SP thymocytes in M1 stage (CD69⁺ MHCI⁺) ($n$ = 6–7 mice per group). (E) Frequency of TCRβ⁺CCR7⁺ SP thymocytes in M2 stage (CD69⁻MHCI⁺) ($n$ = 6–7 mice per group). (F) Frequency of Nur77⁺ cells among DP, CD4SP, or CD8SP thymocytes ($n$ = 6–7 mice per group). CCR, chemokine receptor; CD, cluster of differentiation; cKO, conditional knockout; DN, double negative; DP, double positive; MHCI, major histocompatibility complex class I; M1, mature 1; M2, mature 2; N.S., not significant; Nur77, nuclear receptor 77; pos. sel., positive selection; pre-sel., pre-selection; SM, semimature; SP, single positive; TCR, T cell receptor; WT, wild type; Yap, Yes-associated protein.

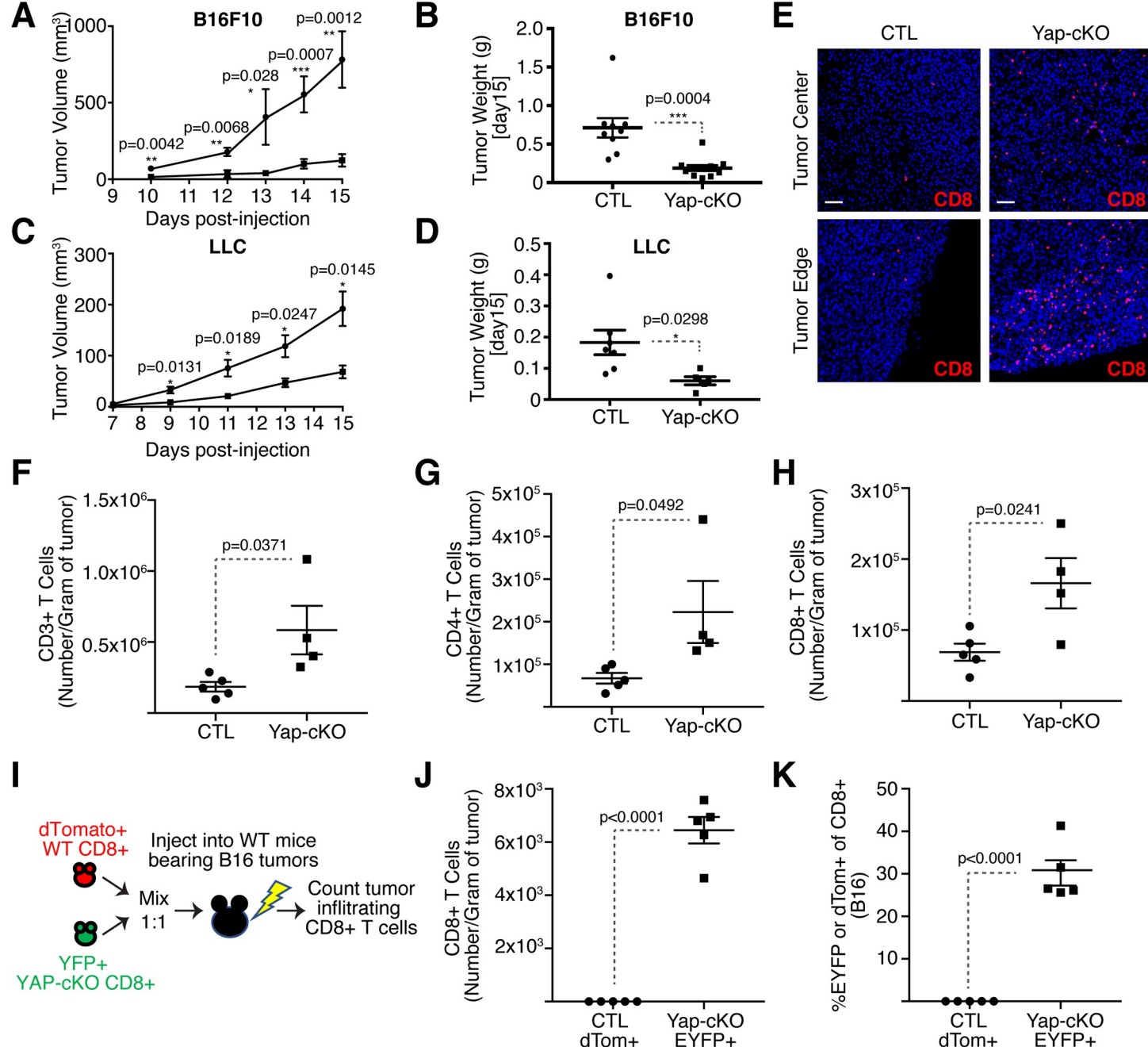

**Fig 4. T-cell–specific deletion of Yap results in reduced tumor growth and enhanced T-cell tumor infiltration.** Mice were challenged subcutaneously with B16F10 or LLC tumor cells on the right flank. Some mice carrying B16F10 tumors received adoptive cell transfer of WT and Yap-cKO CD8⁺ T cells. Tumor growth was monitored over the course of 15 days, until the maximum size of the tumors reached 500 mm³. B16F10 tumors were harvested for immunofluorescence or flow cytometric analysis. Statistical differences were determined by using a Student $t$ test. The underlying data for the graphs in this figure can be found in S4 Data. (A) B16 tumor growth curve of WT and Yap-cKO mice ($n$ = 9/group). (B) Tumor weight of B16 tumors derived from WT and Yap-cKO mice on day 15 post injection ($n$ = 9/group). (C) LLC tumor growth curve of WT and Yap-cKO mice ($n$ = 7/WT group, $n$ = 5/cKO group). (D) Tumor weight of LLC tumors derived from WT and Yap-cKO mice on day 15 post injection ($n$ = 7/WT group, $n$ = 5/cKO group). (E) CD8⁺ T-cell immunofluorescence on day 15 of B16 tumor growth. (F) Absolute numbers of CD3⁺ TILs from WT and Yap-cKO B16 tumors. Tumors were harvested on day 15; stained with antibodies against CD45, CD3, CD4, and CD8; and analyzed using flow cytometry ($n$ = 5/WT mice, $n$ = 4/ Yap-cKO mice). (G) Absolute numbers of CD4⁺ TILs from WT and Yap-cKO B16 tumors, prepared as in Fig 4D ($n$ = 5/WT mice, $n$ = 4/ Yap-cKO mice). (H) Absolute numbers of CD8+ TILs from WT and Yap-cKO B16 tumors, prepared as in Fig 4D ($n$ = 5/WT mice, $n$ = 4/ Yap-cKO mice). (I) Experimental plan for adoptive cell transfer of WT and Yap-cKO CD8⁺ T cells in WT B16-bearing mice. (J) Absolute numbers of dTom⁺ and EYFP⁺ Yap-cKO versus WT CD8+ T cells in C57BL/6 B16 tumors. WT dTom⁺ CD8⁺ T cells were mixed 1:1 with Yap-cKO EYFP⁺ CD8⁺ T cells prior to being injected into WT C57BL/6 mice. Subsequently, mice were injected subcutaneously with B16F10 melanoma cells, and absolute number of infiltrating T cells was determined on day 15 by flow cytometry ($n$ = 5/group). (K) Percentage of dTom⁺ and EYFP⁺ CD8⁺ T cells out of total B16 tumor-infiltrating CD8⁺ T cells ($n$ = 5/group). CD, cluster of differentiation; cKO, conditional knockout; CTL, Control; dTom, dTomato; EYFP, enhanced yellow fluorescent protein; LLC, Lewis lung carcinoma; TIL, tumor-infiltrating lymphocyte; WT, wild type; Yap, Yes-associated protein.

were then measured in tumors after 15 days. Yap-cKO CD8$^+$ T cells showed a significantly enhanced capacity to infiltrate tumors (Fig 4J), with nearly 30% of all CD8$^+$ tumor-infiltrating T cells being EYFP$^+$ Yap-cKO T cells compared to almost undetectable numbers of tdTomato$^+$ WT T cells (Fig 4K). These data are the first to conclusively show that Yap-cKO CD8$^+$ T cells have intrinsically enhanced tumor infiltration capacity.

## Yap regulates global T-cell responses in the local tumor microenvironment

We next aimed to define Yap-regulated signaling networks in T cells that impacted tumor growth and T-cell infiltration. CD4$^+$ and CD8$^+$ TILs were isolated from B16 tumors grown in WT or Yap-cKO mice, as well as CD4$^+$ and CD8$^+$ T cells from corresponding TDLNs and analyzed by RNA-seq (S1–S3 Tables). A large number of genes were differentially expressed in CD4$^+$ and CD8$^+$ TILs isolated from Yap-cKO mice compared to WT mice (Fig 5A and 5B and S2A and S2B Fig). Notably, T cells from TDLNs showed markedly fewer gene expression changes with Yap deletion compared to TILs (Fig 5C and 5D). These data are consistent with Yap function being coordinated with T-cell activation and suggest that the enhanced antitumor responses observed in Yap-cKO T cells are mediated by changes in cellular responses to local tumor signals, including TCR signaling and the cytokine milieu.

Hyper-enrichment analyses [63] of gene expression changes identified in TILs from Yap-cKO mice showed induction of genes related to T-cell activation, differentiation, survival, and migration (Fig 5E and 5F) (S3 Table), implicating Yap in fundamental T-cell processes beyond those previously reported. Yap-cKO TILs showed enrichment of genes associated with TCR signaling, genes that encode major costimulatory molecules, and genes downstream of TCR signaling (S3A and S3B Fig), suggesting enhanced responsiveness to TCR signals in line with our in vitro observations. Cytokine and cytokine receptor signaling gene sets were also enriched in Yap-cKO TILs, suggesting that cytokine production and responsiveness to cytokine receptor engagement were enhanced. Furthermore, chemokine and chemokine receptor signaling gene sets were enriched in Yap-cKO TILs (S3C–S3F Fig), which likely contributes to their improved tumor-infiltrating capacities. Subset-defining transcription factors and cytokines associated with each of the major CD4$^+$ T-cell phenotypes were all up-regulated in Yap KO CD4$^+$ TILs (S3G and S3H Fig), suggesting that Yap deletion leads to enhanced naïve CD4$^+$ T-cell differentiation, consistent with our in vitro observations. Using unique up-regulated genes after differentiation to each of the major T helper subsets [64], our data showed that Yap-cKO CD4$^+$ TILs are more skewed towards a Th2 and Treg phenotype compared to WT CD4$^+$ TILs (S4A–S4D Fig), consistent with prior studies showing that the B16F10 tumor microenvironment enhances these fates [65–67]. As expected, gene set enrichment profiles identified in Yap-cKO TILs also included "Yap1 and TAZ stimulated gene expression" and "signaling by Hippo," which were repressed in both CD4$^+$ and CD8$^+$ TILs (Fig 5G and 5H). An unbiased analysis of transcription factor binding motifs in upstream regulatory regions of the genes altered in Yap-cKO CD4$^+$ and CD8$^+$ TILs revealed the TEAD transcription factor motif as the most significantly enriched in both cell populations (S5A and S5B Fig), suggesting that Yap-regulated TEAD activity contributes to transcriptional regulation of these genes.

Comparison of gene expression changes identified in CD4$^+$ and CD8$^+$ Yap-cKO TILs with clinical data from TCGA [68] showed significant correlation of gene signatures with T-cell infiltration across a variety of human cancers (Fig 5I). Consistent with our preclinical results in melanoma (B16F10) and lung cancer (LLC), we observed significant correlation with T-cell infiltration in skin cutaneous melanoma (SKCM), lung squamous cell carcinoma (LUSC), and lung adenocarcinoma (LUAD). Genes altered in Yap-cKO TILs were also significantly associated with patient survival across several cancers (Fig 4J), as seen most significantly in LUAD

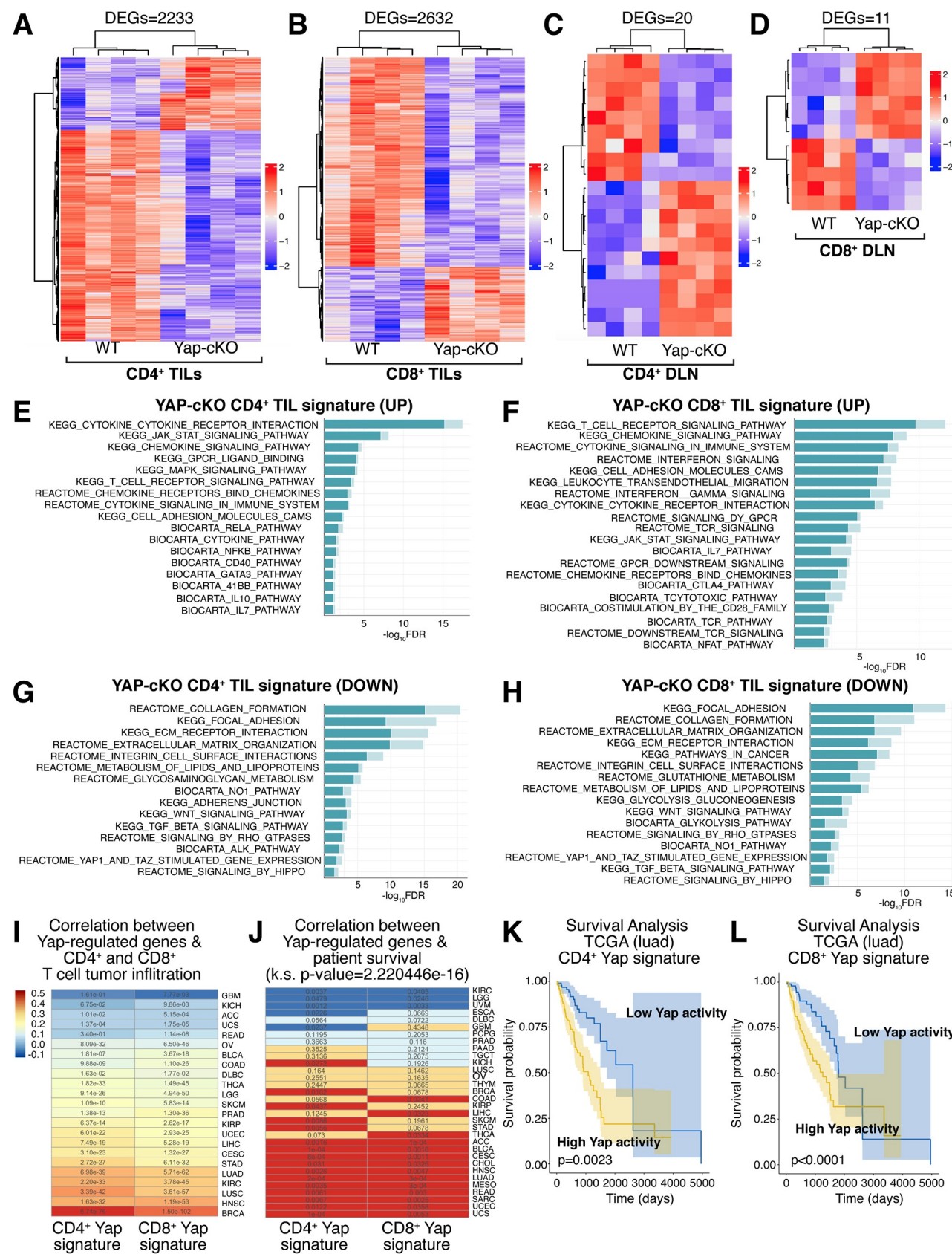

**Fig 5. RNA-seq analysis of Yap-cKO CD4+ and CD8+ B16 TILs uncovers distinct gene expression changes that correlate with T-cell tumor infiltration.** CD4+ and CD8+ TILs and TDLNs were isolated from WT and Yap-cKO mice challenged with B16F10 tumors, and gene expression was analyzed by RNA-seq in the respective cells. RNA-seq data can be found at the NCBI GEO (Series Accession Number GSE139883) and in S1 and S2 Tables. The hyper-enrichment results are outlined in S3 Table. (A) Heatmap showing DEGs identified from Yap-cKO versus WT CD4+ TILs. (B) Yap-cKO versus WT CD8+ TIL DEG heatmap. (C) Yap-cKO versus WT CD4+ TDLN DEG heatmap. (D) Yap-cKO versus WT CD8+ TDLN DEG heatmap. (E) Hyper-enrichment analysis shows enrichment of induced gene sets observed in CD4+ Yap-cKO TILs. (F) Hyper-enrichment analysis shows enrichment of induced gene sets observed in CD8+ Yap-cKO TILs. (G) Hyper-enrichment analysis shows enrichment of repressed gene sets observed in CD4+ Yap-cKO TILs. (H) Hyper-enrichment analysis shows enrichment of repressed gene sets observed in CD8+ Yap-cKO TILs. (I) Gene expression changes identified in Yap-cKO CD4+ and CD8+ TILs correlate with genes reflecting tumor infiltration across many cancers in TCGA data. The heatmap is colored by the coefficient, and the text of each cell represents the adjusted p-value of the correlation. (J) Gene expression changes identified in Yap-cKO CD4+ and CD8+ T cells correlate with patient survival data available in TCGA across several cancers. Red: the average survival probability is higher for patients with high activity of the signature. Blue: the average survival probability is higher for patients with low activity of the signature. Each cell includes the p-value for the survival estimation. The distribution of p-values arising from the multiple survival analyses for each signature across TCGA datasets was compared to a uniform distribution using a Kolmogorov-Smirnov test. (K) Kaplan-Meier survival analysis showing the average survival probability of patients with LUAD that show low versus high Yap activity derived from the Yap-cKO CD4+ gene expression signature. (L) Kaplan-Meier survival analysis showing the average survival probability of patients with LUAD that show low versus high Yap activity derived from the Yap-cKO CD8+ gene expression signature. ACC, adrenocortical carcinoma; BLCA, bladder urothelial carcinoma; BRCA, breast invasive carcinoma; CD, cluster of differentiation; CESC, cervical squamous cell carcinoma and endocervical adenocarcinoma; CHOL, cholangiocarcinoma; cKO, conditional knockout; COAD, colon adenocarcinoma; DEG, differentially expressed gene; DLBC, lymphoid neoplasm diffuse large B-cell lymphoma; DLN, draining lymph node; ESCA, esophageal carcinoma; GBM, glioblastoma multiforme; GEO, Gene Expression Omnibus; HNSC, head and neck squamous cell carcinoma; KICH, kidney chromophobe; KIRC, kidney renal clear cell carcinoma; KIRP, kidney renal papillary cell carcinoma; k.s., Kolmogorov-Smirnov; LGG, brain lower grade glioma; LIHC, liver hepatocellular carcinoma; LUAD, lung adenocarcinoma; LUSC, lung squamous cell carcinoma; MESO, mesothelioma; NCBI, National Center for Biotechnology Information; OV, ovarian serous cystadenocarcinoma; PAAD, pancreatic adenocarcinoma; PCPG, pheochromocytoma and paraganglioma; PRAD, prostate adenocarcinoma; READ, rectum adenocarcinoma; RNA-seq, RNA sequencing; SARC, sarcoma; SKCM, skin cutaneous melanoma; STAD, stomach adenocarcinoma; TCGA, The Cancer Genome Atlas; TDLN, tumor-draining lymph node; TGCT, testicular germ cell tumors; THCA, thyroid carcinoma; THYM, thymoma; TIL, tumor-infiltrating lymphocyte; UCEC, uterine corpus endometrial carcinoma; UCS, uterine carcinosarcoma; UVM, uveal melanoma; WT, wild type; Yap, Yes-associated protein.

for both CD4+ and CD8+ Yap-cKO gene signatures (Fig 4K and 4L). These analyses showed that increased Yap activity in TILs correlated with poorer prognosis and lower T-cell infiltration, suggesting that Yap function in T cells contributes to aggressive human cancer development and cancer immunosuppression.

## Discussion

We present evidence for an inhibitory role for Yap in both CD4+ and CD8+ T cells and offer the first data showing that Yap is a negative regulator of T-cell tumor infiltration. We found that disrupting Yap activity leads to enhanced T-cell activation, augmented differentiation, and increased tumor infiltration. Two major signals are necessary for T-cell activation: signaling from engagement of the TCR with its cognate antigen:MHC complex and a second signal from the costimulatory receptor CD28 binding to ligands CD80 and CD86 [69,70]. Experimental data indicated that CD4+ and CD8+ T cells from Yap-cKO mice were more sensitive to TCR signaling compared to WT cells. We demonstrated significant up-regulation in sensitivity of activation markers CD44, CD25, and CD69 to TCR signal strength. Through recruitment of key effector kinases, phosphatases, and scaffold proteins, transcriptional programs are induced by TCR signaling that leads to production of cytokines and cytokine receptors, which include IL-2 and CD25, the α chain of the heterotrimeric high-affinity IL-2 receptor. IL-2 binding to the IL-2 receptor, together with TCR signaling and costimulation, elicit transcriptional changes resulting in proliferation and differentiation [71]. RNA-seq analysis of Yap-cKO versus WT TILs revealed that CD3, CD28, CD80, and CD86 receptor molecules, kinases, phosphatases, and scaffold proteins are all up-regulated with Yap deletion, offering a mechanism for enhanced activation of CD4+ and CD8+ T cells. These mechanisms are significantly broader in scope than the TGFβ-specific mechanisms previously reported for Yap in Tregs [37], implicating Yap as a regulator of all activated T-cell responses.

Our analysis of thymic populations provides the first evidence that Yap deletion does not cause significant changes in thymocyte maturation. These data suggest that Yap functions primarily in mature T cells, and loss of Yap is not deleterious to thymocytes. While the precise reason for differential action of Yap in thymocytes versus mature T cells remains undefined, other proteins have distinct TCR-induced mechanisms in immature thymocytes compared to mature T cells. For example, Nur77 is less pro-apoptotic in mature T cells compared to thymocytes due to differences in TCR-induced phosphorylation and kinase activity [72]. We observed a slight increase in total number of TCRβ+CD69+ thymocytes in Yap-cKO mice compared to WT mice. This increase in TCR signaling post positive selection is consistent with our observations that Yap enhances T-cell sensitivity to TCR signaling. Interestingly, we did not see a corresponding increase in numbers of Yap-cKO T cells at later maturation stages, suggesting that the increased numbers of positively selected thymocytes may not survive negative selection. Overall, these data suggest that Yap-cKO mice are developmentally similar to WT mice. However, our studies do not rule out the potential that Yap may act by subtly shifting the specificity of the T-cell repertoire during development, which may contribute to some of the effects observed in our work. This remains an important area for future studies.

We demonstrated that Yap-cKO enhanced CD4+ and CD8+ T-cell activation in vitro. Observed increases in TEAD1 and TEAD3 expression in activated T cells suggest that Yap may exert immunosuppressive effects through TEAD-regulated transcription. Consistent with this idea, TEAD-binding motifs were the most enriched motif in the Yap-regulated gene expression signature identified in Yap-cKO TILs. Furthermore, treatment with verteporfin, a reported inhibitor of Yap-TEAD activity [73], increased T-cell activation similar to Yap deletion. Interestingly, neither Yap deletion nor verteporfin treatment significantly impacted T-cell proliferation. Previous studies have demonstrated that proliferation has a sharp, switch-like threshold for TCR signal to elicit proliferation [74], and thus it is likely that our experimental conditions provided optimal signaling above this threshold. Expression levels of activation markers have been directly linked to TCR signal strength, such as CD69 levels being directly regulated by affinity and dose of TCR ligand [75] and CD71 levels being directly linked to mammalian target of rapamycin (mTOR) activation level [76]. Our observations therefore support a novel conceptual advance, which is that Yap may link TCR signal strength to negative feedback.

The rapid induction of Yap post T-cell activation implicates shared canonical signals of T-cell activation in the regulation of Yap. These observations are consistent with those made previously in naïve CD8+ T cells [27] but contrast those arguing Treg-specific roles for Yap [37]. Our data suggest that Yap promotes a normal negative feedback mechanism during T-cell activation similar to inhibitory checkpoint molecules, and inhibition of Yap must be timed before or during T-cell activation. Our observations interestingly also indicate that Yap plays a prominent role only following activating signals from the microenvironment. This is highlighted by the large number of genes impacted in Yap-deleted CD4+ and CD8+ T cells isolated from TILs compared to few genes differentially expressed in TDLNs, and it is consistent with our data showing that Yap levels increase after T-cell activation. These data suggest that therapeutic inhibition of Yap in T cells may have fewer side effects compared to strategies such as checkpoint blockade, since the regulatory activity of Yap is synchronized with T-cell activation and appears specific to sites of active T-cell priming. Yap connects a variety of extracellular stimuli into intracellular cues that inform the cell of its own structural features (actin cytoskeleton, polarity, cell shape) as well as its location and surroundings (mechanical, adhesion, extracellular matrix), instructing cellular survival, proliferation, differentiation, and fate. Therefore, a better understanding of how Yap is regulated by these signals in T cells may reveal new insights into immunoregulatory mechanisms.

Regulation of T-cell activation by the microenvironment through polarizing cytokines allows for diverse, context-specific differentiation and functional diversity in CD4$^+$ T cells [3]. We observe that Yap-cKO CD4$^+$ T cells have increased capacity to differentiate towards Th1, Th2, Th17, and Treg phenotypes. These observations suggest that Yap does not preferentially bias control of T-cell differentiation but instead enhances responsiveness to local microenvironment cues. These data therefore suggest that Yap inhibition may enhance responses to current immunotherapy strategies. Clues into Yap-regulated events are embedded in our RNA-seq of Yap-cKO TILs, which show broad changes in T-cell responses. Genes regulated by Yap include several key effectors of T-cell activation, including genes involved in nuclear factor kappa-light-chain-enhancer of activated B cells (NFκB), mitogen-activated protein kinase (MAPK), nuclear factor of activated T cells (NFAT), and Janus kinase (JAK)-signal transducers and activators of transcription (STAT) signaling. Notably, Yap-regulated genes are also enriched for those regulated by the TGFβ and Wnt pathways, which have important pleotropic roles in T-cell biology [77,78]. Given the known convergence of Yap with these immunomodulatory pathways [79], it is likely that Yap directs their transcriptional targets and signal strength. Our study provides novel evidence of CD4$^+$ and CD8$^+$ T-cell–intrinsic effects of Yap on T-cell activation and tumor infiltration.

An important finding of our studies is that Yap-cKO mice show delayed tumor growth in conjunction with increased infiltration and expression of effector molecules. The well-characterized B16F10 melanoma and LLC lung cancer models generate tumors that are characterized as immune deserts [54–57,80–82]. In immune deserts, T cells are completely excluded from the tumor microenvironment due to suppressed tumor immunogenicity and insufficient T-cell priming, costimulation, and activation [24]. The ability for Yap-cKO CD4$^+$ and CD8$^+$ T cells to become activated and infiltrate tumors that normally inhibit T-cell infiltration is therefore remarkable and important, as these phenotypes are key for combating tumor growth in human cancer patients [83]. Adoptive transfer of Yap-cKO CD8$^+$ T cells showed that CD8$^+$ T cells have intrinsically enhanced tumor infiltration capacity compared to WT host CD8$^+$ T cells into B16F10 tumors. These data are novel and suggest that one of the early events in Yap-mediated tumor immunosuppression may be exclusion of CD8$^+$ T cells from the tumor. The significant correlation of CD8$^+$ and CD4$^+$ Yap-cKO gene signatures with tumor T-cell infiltration in TCGA data suggest that Yap represses CD8$^+$ and CD4$^+$ T-cell migration and tumor infiltration in human cancers. In particular, our preclinical results in melanoma and lung cancer models were mirrored in human SKCM, LUAD, and LUSC. Low activity of Yap-regulated CD4$^+$ and CD8$^+$ gene expression signatures correlated with immune infiltration in 19 TCGA cancer types—and strongly correlated with survival in 11 TCGA cancer types—highlighting that Yap activity in T cells has broad potential as a target for cancer therapy.

Collectively, our data show that Yap is an important immunosuppressor and suggest that inhibition of Yap activity in T cells could have important clinical implications in T-cell therapies against cancer and other diseases. We integrate findings from prior studies and extend them, showing for the first time that Yap is expressed in activated CD4$^+$ and CD8$^+$ T cells and plays a regulatory role in T-cell activation for both subsets. Insights from our study therefore motivate future investigation of Yap function in T-cell homeostasis and disease.

## Materials and methods

### Ethics statement

This study was performed in strict accordance with the recommendations in the Guide for the Care and Use of Laboratory Animals of the National Institutes of Health. Animal care and handling was consistent with the recommendations of the Panel on Euthanasia of the

American Veterinary Medical Association. Prior to the initiation of experiments, all study protocols were reviewed and modified according to the suggestions of the Boston University School of Medicine and the University of Maryland, Baltimore County (UMBC), Institutional Animal Care and Use Committee (IACUC). The Boston University School of Medicine and UMBC animal management programs are accredited by the American Association for the Accreditation of Laboratory Animal Care and meets National Institutes of Health standards as set forth in the Guide for the Care and Use of Laboratory Animals (DHHS Pub. No. [NIH] 85–23, rev 1985). Boston University's Animal Welfare Assurance number is A-3316-01 and UMBC's Animal Welfare Assurance number is D16-00462.

## Mouse strains and genotyping

Yap-loxP/loxP mice, provided by Dr. Jeff Wrana and previously described [84], were backcrossed to the C57BL/6 background for 10 generations and bred with the Tg(Cd4-cre)1Cwi (Jax: 022071) [85] and LSL-EYFP (Jax: 006148) [86] lines to derive Yap-loxP/loxP; LSL-EYFP; CD4-Cre mice. For adoptive cell transfers, CD4-cre mice were crossed with LSL-tdTomato mice (Jax: 007914) [87]. All experiments were performed with 6- to 10-week-old mice, and littermates were always used as controls for each experiment. Animal protocols and study designs were approved by Boston University School of Medicine and UMBC. Mice were maintained in pathogen-free facilities at BUMC and UMBC and were PCR genotyped using published protocols [84–87].

## Cell culture and mouse tumor challenges

B16F10 mouse melanoma cells (ATCC CRL-6475) and LLC1 cells (ATCC CRL-1642) were cultured in DMEM supplemented with glucose, L-glutamine, sodium pyruvate, 10% FBS, penicillin, and streptomycin. Cells were split once they reached 70% confluency and were not used for mouse challenge past a fifth passage. T cells were cultured in RPMI 1640 supplemented with 10% FBS, 1 mM sodium pyruvate, 50 μM β-ME, penicillin, streptomycin, 2 mM L-glutamine, 100 mM nonessential amino acids, 5 mM HEPES, free acid and β-mercaptoethanol. For tumor inoculations, $5 \times 10^4$ B16F10 cells or $5 \times 10^5$ LLC cells were injected subcutaneously on the right flank of each mouse on day 0. Tumor volume was estimated using the formula $(L \times W^2) \div 2$. Survival endpoint was reached once the tumors measured 500 mm$^3$, around day 15. Mice were euthanized by isoflurane inhalation and subsequent cervical dislocation, and tumors were harvested for further prospecting.

## T-cell isolation

Spleens from WT or Yap-cKO mice were pushed through a 70 μm mesh (Falcon) using an insulin syringe plunger and washed with PBS. Cells were treated with ACK red blood cell lysis buffer, and splenocyte single-cell suspensions were prepared for magnetic separation or stained for sorting by flow cytometry. CD4$^+$ or CD8$^+$ T-cell enrichment was performed using magnetic beads (Miltenyi Biotec or STEMCELL Technologies). Naïve CD4$^+$ T cells were isolated using a naïve CD4$^+$ T-cell isolation kit (Miltenyi Biotec).

## Flow cytometry

Isolated splenocytes or tumor digests were washed with PBS and stained with the LIVE/DEAD fixable near-IR dead cell stain kit (Invitrogen). Cells were then washed with stain buffer (BD), resuspended in stain buffer containing Fc block (BD), and incubated for 5 minutes at 4˚C. Surface antibodies were added in predetermined concentrations, and cells were incubated for 30 minutes at 4˚C or 15 minutes at room temperature, before being washed with BD stain buffer

and resuspended in PBS for flow cytometric analysis. For intracellular cytokine staining, cells were fixed and permeabilized using the BD Cytofix/Cytoperm Fixation/Permeabilization kit (BD) after dead cell dye and surface staining. For transcription factor staining, the eBioscience Foxp3/transcription factor staining buffer set was used. Flow cytometry analyses were performed on BD LSRII at Boston University School of Medicine Flow Cytometry Facility or at the University of Maryland School of Medicine Center for Innovative Biomedical Resources, Flow Cytometry Shared Service and analyzed by FlowJo (TreeStar).

### T-cell activation and proliferation assays

Cells were cultured and stimulated in 96-well plates at $1 \times 10^5$ cells per well. Plates were coated with anti-CD3 antibody (Biolegend) at concentrations of 0, 0.125, 0.25, 0.5, and 1 μg/ml at 4˚C overnight and were washed twice with PBS before incubation. Cells were stimulated in the anti-CD3–coated plates with soluble anti-CD28 at 2 μg/ml (Biolegend). On days 1 and 3, cells were stained with dead cell dye as well as antibodies recognizing the lineage and activation markers CD3 BUV737 (BD), CD4 BUV395 (BD), CD8 PerCP-Cy5.5 (Biolegend), CD69 PE (Biolegend), CD44 BV650 (Biolegend), and CD25 APC (Biolegend). For proliferation assays, CD4$^+$ or CD8$^+$ T cells were isolated and stained using the CellTrace Violet or CFSE Cell Proliferation Kit (Life Technologies). Briefly, purified cells were washed with PBS and incubated with CellTrace dye for 20 minutes at 37˚C protected from light. After 20 minutes, complete RMPI medium was added to the cell suspension, and the cells were incubated 5 minutes further before being washed and resuspended in complete RPMI medium. Cells were cultured in 96-well plates at $1 \times 10^5$ cells per well and were stimulated using anti-CD3/CD28 dynabeads (Gibco) at a 1:1 ratio with T cells. On days 1 and 3, cells were stained with dead cell dye, and proliferation was measured at the same time.

### CD4$^+$ T-cell in vitro differentiation

For CD4$^+$ T-cell in vitro differentiation into Th1, Th2, Th17, and Treg subsets, naïve CD4$^+$ T cells were enriched using magnetic beads (Miltenyi Biotec). Purified cells were plated at $1 \times 10^5$ cells per well on a 96-well plate coated with 10 μg/ml anti-CD3 (Biolegend) and cultured with 2 μg/ml soluble anti-CD28 (Biolegend). The following conditions were specific to each differentiation regime: Th1: 10 ng/ml IL-12 and 10 μg/ml anti-IL-4; Th2: 50 ng/ml IL-4, 10 μg/ml anti-IFNγ, and 10 μg/ml anti-IL-12; Th17: 50 ng/ml IL-6, 20 ng/ml IL-1β, 5 ng/ml IL-23, 1 ng/ml TGFβ, 12 μg/ml αIFNγ, and 10 μg/ml anti-IL-4; and Treg: 100 IU/ml IL2, and 0, 0.5, or 5 ng/ml TGFβ. Cytokines and antibodies were purchased from Biolegend, except for IL-2 and TGFβ (R&D). Cells were cultured for 5 days, before being stimulated with 50 ng/ml PMA (Sigma, P1585) and 1 μg/ml ionomycin (Sigma, I0634) for 6 hours at 37˚C in the presence of Golgistop (monensin, BD) or Golgiplug (brefeldin, BD) added after the first 30 minutes of stimulation. Cells were stained with the LIVE/DEAD fixable near-IR dead cell stain kit (Invitrogen); antibodies for surface markers CD3 BUV737 (BD), CD4 BUV395 (BD), CD8 APCFire750 (Biolegend), and CD25 BV510 (Biolegend); antibodies for intracellular cytokines IFNγ APC (Biolegend) and IL-17 PerCP-Cy5.5 (Biolegend); or antibodies for transcription factors GATA3 PECy7 (Biolegend) and Foxp3 PE (BD), as described earlier.

### Thymocyte phenotyping

Thymuses from WT or Yap-cKO mice (8–10 weeks old) were mechanically disrupted by being pushed through a 70 μm mesh (Falcon) with an insulin syringe plunger and washed with PBS. Cells were treated with ACK red blood cell lysis buffer, and thymocyte single-cell suspensions were stained for analysis by flow cytometry as described earlier. Cells were stained with dead

cell dye and antibodies for the following surface markers: CD3 BUV737 (BD), CD4 BUV395 (BD), CD8 PerCP-Cy5.5 (Biolegend), TCRβ BV510 (Biolegend), CCR7 PECy7 (Biolegend), H-2Kb PE (Biolegend), CD69 BV421 (Biolegend), CD45R/B220 APCFire750 (Biolegend), CD25 APCFire750 (Biolegend), GL3 APCFire750 (Biolegend), and NK1.1 APCFire750 (Biolegend). Thymocytes were also stained intracellularly with anti-Nur77 APC (BD) using the eBioscience Foxp3/Transcription factor staining buffer set, as described earlier.

### Tumor digestion

B16 tumors from WT and Yap-cKO mice were dissected, mechanically disrupted, and digested in serum-free media containing 2 mg/ml collagenase type I (Worthington) and DNase I (Sigma) for 30 minutes at 37˚C in a rotator. Tumor digests were then passed through a 70 μm mesh (Falcon) using an insulin syringe plunger and washed with PBS. Cells were treated with ACK red blood cell lysis buffer (Gibco), and tumor single-cell suspensions were prepared for staining and analysis by flow cytometry, using DAPI (Biolegend) and antibodies for CD45 BV510 (Biolegend), CD3 BUV737 (BD), CD4 BUV395 (BD), and CD8 PerCP-Cy5.5 (Biolegend). For determining absolute numbers of tumor-infiltrating T cells by flow cytometry, AccuCount fluorescent particles (Spherotech) were added to the tumor digests.

### Adoptive cell transfers

For the adoptive cell transfers, EYFP[+] Yap-cKO CD8[+] T cells were mixed 1:1 with tdTomato[+] WT CD8[+] T cells and injected intravenously into 8-week-old WT C57BL/6 mice (Taconic) on day 0. Mice also received a subcutaneous injection of $5 \times 10^4$ B16F10 cells on the same day. On day 15 of tumor growth, tumors were harvested and stained for flow cytometric analysis with dead cell dye, DAPI (Biolegend), and antibodies recognizing CD45 BV510 (Biolegend), CD3 BUV737 (BD), CD4 APC (Biolegend), and CD8 PerCP-Cy5.5 (Biolegend).

### Immunofluorescence microscopy

Harvested B16 tumors were fixed overnight in PLP fixative, followed by incubation in 15% and 30% sucrose. Tumors were embedded in OCT and frozen. Cryosections were cut at 5 μm thickness and stored at −20˚C. Slides were stained with rat anti-mouse CD8 (clone CT-CD8a, Fisher) and donkey anti-rat Alexa 647 (Jackson Immuno Research Labs). Slides were washed and mounted in ProLong antifade reagent with DAPI (Life Technologies). Images were acquired using an AxioObserver D1 equipped with an X-Cite 120LED System.

### Immunoblotting and quantitative real-time PCR

RNA was extracted using Rneasy Mini Kit (Qiagen), and 1 μg was used to generate cDNA using an iScript cDNA Synthesis Kit (Bio-rad). Taqman primers (Life Tech) for mouse Gapdh (4352339E), Yap (Mm01143263_m1), or TEAD1-4 (Mm00493507_m1, Mm00449004_m1, Mm00449013_m1, Mm01189836_m1) were mixed with cDNA and Taqman Universal Master Mix II (Life Tech), and ddC$_T$ values were calculated relative to unstimulated controls. Protein lysates were analyzed by immunoblotting with anti-YAP (D8H1X) XP (CST 14074) and anti-GAPDH (D16H11) XP (CST 8884) antibodies (original immunoblot images as shown in S5 Data) and were imaged using a Bio-Rad ChemiDoc system.

### Sample preparation for RNA-seq

B16 tumors from WT and Yap-cKO mice were digested as described earlier. Tumors cells were subsequently stained with DAPI and antibodies recognizing CD45 V500 (Biolegend),

CD3 PE (Biolegend), CD4 APC (Biolegend), CD8 PerCP-Cy5.5 (Biolegend), and CD4[+] and CD8[+] TILs were sorted from each tumor using a BD FACSAria instrument. CD4[+] and CD8[+] T cells were sorted from WT and Yap-cKO TDLNs, as well. Cells were sorted into TRIzol LS reagent (Invitrogen), and RNA was isolated using a miRNeasy micro kit (Qiagen).

### Transcriptomic analyses and gene expression signature extraction

RNA quality was evaluated using Agilent Bioanalyzer 2100 Eukaryote Total RNA Pico chips. RNA-seq libraries were prepared using the SMART-Seq version 4 Ultra Low Input RNA Kit (Takara, 634889) from total RNA, following the manufacturer's protocol. Libraries were then sequenced on a HiSeq 4000 using 75-bp paired end reads to an average depth of 22,445,650 ± 240,398 reads (SEM). Transcript abundance estimates were quantified using Salmon to mouse reference transcriptome from assembly GRCm38 (mm10), aggregated to gene level for UCSC-annotated genes using tximport, and DESeq2 was used to calculate normalized counts and differential expression [88, 89]. CD4 and CD8 up/down gene signatures were generated through differential expression analysis via DESeq2. Differentially expressed genes (DEGs) between Yap-cKO versus WT CD4[+] and CD8[+] cells were defined as log2(FC) > 1 (up) or log2(FC) < −1 (down) and FDR < 0.05. DEGs were visualized with a heatmap combined with a barplot annotation, with the heatmap cells representing the log-normalized expression values for each sample. Each row is accompanied by a bar representing the log-fold change in gene expression (KO/WT). Plots were generated using the *ComplexHeatmap* software package available in R. Data have been deposited in NCBI GEO (accession number GSE139883).

### Analysis of gene expression signatures in TCGA datasets

The activation of CD4/8 up and down signatures was calculated with Gene Set Variation Analysis (GSVA) [90] in primary tumor samples across multiple TCGA RNA-seq datasets. Signature activation was summarized by the sum of activation of the up-regulated signature and inactivation of the down-regulated signature. For example, $CD4_{sig}$ activity = GSVA($CD4_{up\ sig}$) GSVA($CD4_{down\ sig}$). Heatmaps for all TCGA analyses were generated with the *pheatmap* software package available in R. TCGA data include count matrices generated with STAR2/HTSeq downloaded from Genomic Data Commons (GDC) Data Portal. Each count matrix was normalized by relative log expression (RLE) with DESeq2. This analysis was constrained to samples for which survival information was available and immune infiltration could be estimated with Tumor Immune Estimation Resource (TIMER) [91]. The average T-cell infiltration per sample was estimated with TIMER in each TCGA dataset, represented as a single heatmap cell. This average was measured separately for CD4 T-cell and CD8 T-cell infiltration. Additionally, these values were summed to observe their additive effect. Transcription factor motif analysis was performed using the HOMER de novo motif analysis tool [92].

For correlation of signatures with T-cell infiltration, CD4/8 signature activation was correlated with the sum of CD4[+] T-cell and CD8[+] T-cell infiltration in each TCGA dataset estimated by TIMER [91]. The heatmap in Fig 5I is colored by the correlation coefficient, and the text of each cell is the adjusted *p*-value of the correlation. For survival analysis, the CD4/8 signature activation was used to stratify patients across TCGA datasets into high or low activated groups separated by the mean. For each dataset, Kaplan-Meier survival plots were generated, and the results were summarized with the heatmap shown in Fig 5J. Red signifies that an average survival probability is higher for patients with high activity of Yap-regulated signature, whereas dark blue signifies that an average survival probability is higher for patients with low Yap activity. Light orange and light blue signify the same groups with higher survival probability, respectively, but the differences are not significant. The text of each cell is the *p*-value for the survival estimation. The distribution of *p*-values arising from the multiple survival analyses

 

for each signature across TCGA datasets was compared to a uniform distribution using a Kolmogorov-Smirnov test.

## Supporting information

**S1 Fig. Yap deletion or pharmacological inhibition of Yap does not significantly affect T-cell proliferation.** CD4$^+$ and CD8$^+$ T cells were isolated from WT or Yap-cKO mice and screened for EYFP expression as well as activation marker expression and proliferation after αCD3 and αCD28 stimulation. Proliferation was also tested for WT CD4$^+$ and CD8$^+$ T cells treated with increasing concentrations of verteporfin under IL-2, anti-CD3, and anti-CD28 stimulation. Statistical differences were determined by using a Student $t$ test, with "ns" indicating not significant. (A) Yap mRNA expression in WT or Yap-cKO CD4$^+$ T cells following CD3/CD28 stimulation. (B) Yap mRNA expression in WT or Yap-cKO CD8$^+$ T cells following CD3/CD28 stimulation. (C) EYFP expression by flow cytometry on CD4$^+$ cells isolated from WT or Yap-cKO mouse spleens. (D) EYFP expression by flow cytometry on CD8$^+$ cells isolated from WT or Yap-cKO mouse spleens. (E) CD69 expression on WT and Yap-cKO CD4$^+$ T cells 72 hours post CD3/CD28 stimulation ($n = 2$–$3$ per dose/group). (F) CD69 expression on WT and Yap-cKO CD8$^+$ T cells 72 hours post CD3/CD28 stimulation ($n = 2$–$3$ per dose/group). (G) WT and Yap-cKO CD4$^+$ T-cell proliferation ($n = 3$/group). (H) WT and Yap-cKO CD8$^+$ T-cell proliferation ($n = 3$/group). (I) CD69 expression on WT CD4$^+$ T cells 72 hours post IL-2 and CD3/CD28 stimulation and increasing concentration of verteporfin ($n = 4$/group). (J) CD69 expression on WT CD4$^+$ T cells 72 hours post IL-2 and CD3/CD28 stimulation and increasing concentration of verteporfin ($n = 4$/group). (K) Proliferation of DMSO-versus verteporfin-treated WT CD4$^+$ and CD8$^+$ T cells (representative of 4 independent experiments). Raw data for this experiment are available in FLOWRepository (Repository ID: FR-FCM-Z2D5).
(TIF)

**S2 Fig. Top 25 up- and down-regulated genes responding to Yap deletion in CD4$^+$ and CD8$^+$ TILs.** RNA-seq was performed from CD4$^+$ and CD8$^+$ TILs and TDLNs that were isolated from WT and Yap-cKO mice challenged with B16F10 tumors (data at NCBI GEO GSE139883 and listed in S1 and S2 Tables), and the top DEGs are shown. (A) A heatmap representing the top and bottom 25 DEGs in Yap-cKO versus WT CD4$^+$ TILs. (B) A heatmap representing the top and bottom 25 DEGs in Yap-cKO versus WT CD8$^+$ TILs.
(TIF)

**S3 Fig. Expression of genes related to T-cell activation, chemokines and chemokine receptors, and T-helper subset–defining factors are up-regulated in Yap-cKO CD4$^+$ and CD8$^+$ TILs.** DEGs identified in Yap-cKO versus WT CD4$^+$ and CD8$^+$ TILs that encode factors related to T-cell function are shown. These data were derived from RNA-seq analysis of the respective mice challenged with B16F10 tumors, which is available at NCBI GEO (GSE139883) and listed in S1 and S2 Tables. (A) Log$_{10}$(normalized RNA-seq counts +1) of T-cell activation–related genes in Yap-cKO versus WT CD8$^+$ TILs. (B) Log$_{10}$(normalized RNA-seq counts +1) of T-cell activation–related genes in Yap-cKO versus WT CD4$^+$ TILs. (C) Log$_{10}$(normalized RNA-seq counts +1) of chemokine genes in Yap-cKO versus WT CD8$^+$ TILs. (D) Log$_{10}$(normalized RNA-seq counts +1) of chemokine receptor genes in Yap-cKO versus WT CD8$^+$ TILs. (E) Log$_{10}$(normalized RNA-seq counts +1) of chemokine genes in Yap-cKO versus WT CD4$^+$ TILs. (F) Log$_{10}$(normalized RNA-seq counts +1) of chemokine receptor genes in Yap-cKO versus WT CD4$^+$ TILs. (G) Log$_{10}$(normalized RNA-seq counts +1) of T-helper subset–defining cytokines in Yap-cKO versus WT CD4$^+$ TILs. (H) Log$_{10}$(normalized RNA-seq counts +1)

of T-helper subset–defining transcription factors in Yap-cKO versus WT CD4$^+$ TILs. Significant differences were determined by a Student $t$ test; $^*p < 0.05$; $^{**}p < 0.01$; $^{***}p < 0.001$. (TIF)

**S4 Fig. Yap-cKO TILs are skewed towards Th2 and Treg gene expression signatures compared to WT.** DEGs identified in Yap-cKO versus WT CD4$^+$ TILs that represent different CD4$^+$ fates are shown. These data were derived from RNA-seq analysis of the respective mice challenged with B16F10 tumors, which is available at NCBI GEO (GSE139883) and listed in S1 and S2 Tables. (A) Heatmap of statistically significant differentially expressed Th1-related genes in Yap-cKO versus WT CD4$^+$ TILs. (B) Heatmap of statistically significant differentially expressed Th2-related genes in Yap-cKO versus WT CD4$^+$ TILs. (C) Heatmap of statistically significant differentially expressed Th17-related genes in Yap-cKO versus WT CD4$^+$ TILs. (D) Heatmap of statistically significant differentially expressed Treg-related genes in Yap-cKO versus WT CD4$^+$ TILs. (TIF)

**S5 Fig. The TEAD-binding motif is enriched in upstream regulatory elements found in genes altered in expression within Yap-deleted TILs.** HOMER de novo motif analysis was performed on down-regulated gene expression changes identified in Yap-cKO versus WT (A) CD4$^+$ and (B) CD8$^+$ TILs, revealing the TEAD transcription factor motifs among the top enriched motifs. (TIF)

**S1 Table. DEGs identified by RNA-seq analyses of Yap-cKO versus WT CD4$^+$ and CD8$^+$ TILs that were isolated from the respective mice challenged with B16F10 tumors.** The RNA-seq data are available at NCBI GEO (GSE139883). (XLSX)

**S2 Table. DEGs identified by RNA-seq analyses of Yap-cKO versus WT CD4$^+$ and CD8$^+$ TDLNs that were isolated from the respective mice challenged with B16F10 tumors.** The RNA-seq data are available at NCBI GEO (GSE139883). (XLSX)

**S3 Table. Hyper-enrichment analysis of the up-regulated and down-regulated genes identified in Yap-cKO versus WT CD4$^+$ and CD8$^+$ TILs.** (XLSX)

**S1 Data. Original data for the graphs in Fig 1.** Each tab includes data for the noted panels in Fig 1. (XLSX)

**S2 Data. Original data for the graphs in Fig 2.** Each tab includes data for the noted panels in Fig 2. (XLSX)

**S3 Data. Original data for the graphs in Fig 3.** Each tab includes data for the noted panels in Fig 3. (XLSX)

**S4 Data. Original data for the graphs in Fig 3.** Each tab includes data for the noted panels in Fig 4. (XLSX)

**S5 Data. Original data for the graphs in S1 Fig.** Each tab includes data for the noted panels in S1 Fig.
(XLSX)

**S6 Data. Original data for the graphs in S3 Fig.** Each tab includes data for the noted panels in S3 Fig.
(XLSX)

**S1 Raw Images. Original immunoblots that are cropped in Fig 1A and 1B.** Yap immunoblots from (A) CD4$^+$ and (B) CD8$^+$ cells, and GAPDH immunoblots from (C) CD4$^+$ and (D) CD8$^+$ cells.
(PDF)

## Acknowledgments

We acknowledge support from the Boston University Flow Cytometry core, especially from Anna Belkina, as well as support from the University of Maryland School of Medicine Center for Innovative Biomedical Resources, Flow Cytometry Shared Service and the Institute for Genome Sciences for RNA sequencing. We also thank Dr. Jeffrey Wrana for sharing the Yap-loxP mice.

## Author Contributions

**Conceptualization:** Eleni Stampouloglou, Gregory L. Szeto, Xaralabos Varelas.

**Data curation:** Eleni Stampouloglou, Nan Cheng, Anthony Federico, Stefano Monti, Gregory L. Szeto.

**Formal analysis:** Eleni Stampouloglou, Nan Cheng, Anthony Federico, Stefano Monti, Gregory L. Szeto.

**Funding acquisition:** Gregory L. Szeto, Xaralabos Varelas.

**Investigation:** Eleni Stampouloglou, Nan Cheng, Emily Slaby, Stefano Monti, Gregory L. Szeto, Xaralabos Varelas.

**Methodology:** Eleni Stampouloglou, Nan Cheng, Anthony Federico, Stefano Monti, Gregory L. Szeto, Xaralabos Varelas.

**Project administration:** Xaralabos Varelas.

**Resources:** Xaralabos Varelas.

**Software:** Anthony Federico.

**Supervision:** Xaralabos Varelas.

**Writing – original draft:** Eleni Stampouloglou, Gregory L. Szeto, Xaralabos Varelas.

**Writing – review & editing:** Eleni Stampouloglou, Gregory L. Szeto, Xaralabos Varelas.

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
