## [Editor Report · Decision Letter 0]

9 Jun 2019

Dear Dr Varelas, 

Thank you for submitting your manuscript entitled "Yap suppresses T cell function and infiltration in the tumor microenvironment" for consideration as a Short Reports by PLOS Biology.

Your manuscript has now been evaluated by the PLOS Biology editorial staff as well as by an academic editor with relevant expertise and I am writing to let you know that we would like to send your submission out for external peer review.

Please re-submit your manuscript within two working days, ie. by Jun 12 2019 11:59PM.

Kind regards,

Ines

--

Ines Alvarez-Garcia, PhD

Senior Editor

PLOS Biology

Carlyle House, Carlyle Road

Cambridge, CB4 3DN

+44 1223–442810

---

## [Decision Letter · Decision Letter 1]

16 Jul 2019

Dear Dr Varelas,

Thank you very much for submitting your manuscript "Yap suppresses T cell function and infiltration in the tumor microenvironment" for consideration as a Short Reports at PLOS Biology. Your manuscript has been evaluated by the PLOS Biology editors, an Academic Editor with relevant expertise, and by two independent reviewers.

In light of the reviews (below), we will not be able to accept the current version of the manuscript, but we would welcome resubmission of a much-revised version that takes into account the reviewers' comments. We will need to see additional data that directly addresses the question raised by reviewer 2: whether the phenotype you observed is due to differences in the TCR selection process in the thymus or the lack of Yap in peripheral T cells. We cannot make any decision about publication until we have seen the revised manuscript and your response to the reviewers' comments. Your revised manuscript is also likely to be sent for further evaluation by the reviewers.

We appreciate that the requested experiment(s) requires a long time to carry out, and we are willing to relax our standard revision time to allow you six months to revise your manuscript.

Your revisions should address the specific points made by each reviewer. Please submit a file detailing your responses to the editorial requests and a point-by-point response to all of the reviewers' comments that indicates the changes you have made to the manuscript. In addition to a clean copy of the manuscript, please upload a 'track-changes' version of your manuscript that specifies the edits made. This should be uploaded as a "Related" file type. You should also cite any additional relevant literature that has been published since the original submission and mention any additional citations in your response. 

Before you revise your manuscript, please review the following PLOS policy and formatting requirements checklist PDF: http://journals.plos.org/plosbiology/s/file?id=9411/plos-biology-formatting-checklist.pdf. It is helpful if you format your revision according to our requirements - should your paper subsequently be accepted, this will save time at the acceptance stage.

Please note that as a condition of publication PLOS' data policy (http://journals.plos.org/plosbiology/s/data-availability) requires that you make available all data used to draw the conclusions arrived at in your manuscript. If you have not already done so, you must include any data used in your manuscript either in appropriate repositories, within the body of the manuscript, or as supporting information (N.B. this includes any numerical values that were used to generate graphs, histograms etc.). For an example see here: http://www.plosbiology.org/article/info%3Adoi%2F10.1371%2Fjournal.pbio.1001908#s5.

For manuscripts submitted on or after 1st July 2019, we require the original, uncropped and minimally adjusted images supporting all blot and gel results reported in an article's figures or Supporting Information files. We will require these files before a manuscript can be accepted so please prepare them now, if you have not already uploaded them. Please carefully read our guidelines for how to prepare and upload this data: https://journals.plos.org/plosbiology/s/figures#loc-blot-and-gel-reporting-requirements.

Upon resubmission, the editors will assess your revision and if the editors and Academic Editor feel that the revised manuscript remains appropriate for the journal, we will send the manuscript for re-review. We aim to consult the same Academic Editor and reviewers for revised manuscripts but may consult others if needed.

When you submit a revised version of your manuscript, please go to https://www.editorialmanager.com/pbiology/ and log in as an Author. Click the link labelled 'Submissions Needing Revision' where you will find your submission record. 

Sincerely,

Di Jiang, PhD

Associate Editor

Ines Alvarez-Garcia, PhD, 

Senior Editor

PLOS Biology

Reviewer remarks:

Reviewer #1: This is an interesting and very well performed study that investigates the role of the central Hippo pathway effector Yap in different T cell subsets. The role of this pathway in blood is underappreciated and, as such, the manuscript is valuable as it reveals new insights into how it functions in T cells. A previous study in Cancer Discovery had reported a role for YAP in T cell immunity and specifically that it was required for Treg differentiation. The present study is distinct from this as they define previously unappreciated roles for Yap in CD8+ T cells and other T cell subsets in the context of cancer immunity. The find that Yap expression is elevated in activated T cells and that Yap deletion causes elevated T-cell activation and differentiation. This gives new insights into the normal role of Hippo/Yap in the immune compartment and, given the authors’ findings that tumor growth is repressed in their Yap KO T cell models, further rationale for the use of YAP inhibitors in human cancers. Mechanistically, they show that Yap-deficient T cells have a greater ability to infiltrate tumors and they explore this further using RNA-seq – again, this is very relevant for the burgeoning field of cancer immunotherapy. They also explore TCGA datasets with their YAP signatures and find correlations with patient outcomes in several different cancers. The manuscript presents a substantial amount of valuable new knowledge and, as such, I think the paper is suitable for PLos Biol. I have the following suggestions for minor revisions to improve the paper.

An obvious question is which transcription factor works with Yap in T cells. Genetic experiments with TEADs would be very challenging, given there are 4 but can the authors make predictions on whether TEADs will be relevant here? For example, they could assess TEAD1-4 expression in the different T cell subsets at different stages of development and/or mine their RNA-seq data for expression of TEADs. Although a correlative rather than definitive experiment, it might also be possible to assess the relative presence of TEAD motifs in their RNA-seq datasets.

Reviewer #2: The author’s present exciting observations that Yap is highly uupregulated in both CD4 and CD8 T cells following stimulation. They then go on to generate a conditional Yap knockout in T cells by crossing Floxed Yap mice with a Cd4-Cre. They find that the CD4 T cells from these mice have a higher capacity to generate Th1/Th2/Th17 cells and YapKO-CD8 T cells had a higher potential to infiltrate tumors and generate an anti-tumor response. These data are therefore very intriguing and have high potential in anti-tumor immune responses.

A few issues should be addressed:

-This study (as well as a previous study from Ni et al., Cancer Discovery 2018; ref 37) assesses the function of Yap-deficient T cells in a conditional knockout which generates thymic progenitors that are knockout for this gene (using the CD4-Cre). As such, mature T cells in the periphery are generated from Yap-KO double positive thymocytes. The problem is that the authors assume that the selection of WT and Yap-KO thymocytes is equivalent (to be fair, this assumption is also made in the Ni et al publication). As this may not be the case, the data presented may be due to differences in the TCR selection process in the thymus and NOT directly to the lack of Yap in peripheral T cells. This issue is not simple to address but could be evaluated by 1) crossing the conditional YapKO mice to a transgenic TCR and evaluating the ability of those T cells to mount an immune response against a tumor bearing that antigen (i.e. OT1 TCR and B16-Ova tumor); 2) knocking down Yap in peripheral T cells from WT mice by an shRNA approach; or 3) introducing Yap (i.e. retroviral transduction) into the KO T cells to assess whether the observed differences are inhibited.

- As the authors point out, their data differ from those published by Ni et al. which reported Treg-specific functions for Yap. Given these differences, what are the kinetics of Yap upregulation (and downregulation) and how does it compare to CD25/CD71? How does the induction of Yap compare in Treg and Th1 effector conditions? Do the authors here also find a decreased ability of Yap-ko naive CD4 T cells to generate Tregs? As the relative induction of Yap in figure 1 appears to be 70 in CD4 T cells and 45 in CD8 T cells; is is induced to higher levels in CD4s?

-The authors state that phenotype of “Yap-cKO CD4+ TILs are more skewed towards a Th2 and Treg phenotype compared to WT CD4+ TILs”. How then do the authors explain the improved anti-tumor effect of these cells? Is this in contradiction with the previous publication that finds decreased iTreg generation and function in Yap-cKO T cells?

---

## [Editor Report · Decision Letter 2]

22 Nov 2019

Dear Dr Varelas,

Thank you for submitting your revised Short Reports entitled "Yap suppresses T cell function and infiltration in the tumor microenvironment" for publication in PLOS Biology. I have now obtained advice from the Academic Editor and discussed these comments with the other editors.

While the revision seems in general satisfactory, the Academic Editor feels that the new experiments do not really address the question of effects of Yap deletion on T cell selection, nor the stage of Yap deletion/TCR specificity. However, in balance we do feel that the notion that Yap does affect the activation strength of T cells and might play a role tumour-reactive T cells in this particular way is interesting. Thus we will probably accept this manuscript for publication, assuming that you will modify the manuscript to add a caveat stating that whether Yap-deletion has an effect on T cell specificity/repertoire which could contribute to the phenotype described in this study remains an important question to be addressed in the future. Please also make sure to address the data and other policy-related requests noted at the end of this email.

We expect to receive your revised manuscript within two weeks. Your revisions should address the specific points made by the editors. In addition to the remaining revisions and before we will be able to formally accept your manuscript and consider it "in press", we also need to ensure that your article conforms to our guidelines. A member of our team will be in touch shortly with a set of requests. As we can't proceed until these requirements are met, your swift response will help prevent delays to publication.

*Copyediting*

*Published Peer Review History*

*Early Version*

*Submitting Your Revision*

Sincerely,

Ines

--

Ines Alvarez-Garcia, PhD

Senior Editor

PLOS Biology

Carlyle House, Carlyle Road

Cambridge, CB4 3DN

+44 1223–442810

DATA POLICY: IMPORTANT, PLEASE READ

Fig. 1A, B, C, D, E, F, G, H, I, J; Fig. 2A, B, C, D; Fig. 3A, B, C, D, E, F; Fig. 4A, B, C, D, F, G, H, J, K; Fig. 5A, B, C, D, E, F, G, H, K, L; Fig. S1A, B, E, F, G, H, I, J; Fig. S2A, B, Fig. S3A, B, C, D, E, F, G, H and Fig. S4A, B, C, D

---

## [Editor Report · Decision Letter 3]

18 Dec 2019

Dear Dr Varelas,

On behalf of my colleagues and the Academic Editor, Avinash Bhandoola, I am pleased to inform you that we will be delighted to publish your Short Reports in PLOS Biology. 

Early Version

PRESS 

Kind regards,

Hannah Harwood

Publication Assistant, 

PLOS Biology

on behalf of

Ines Alvarez-Garcia,

Senior Editor

PLOS Biology